# Prognostic Factors for Invasiveness and Recurrence of Pituitary Adenomas: A Series of 94 Patients

**DOI:** 10.3390/diagnostics12102413

**Published:** 2022-10-05

**Authors:** Petros Stefanidis, Georgios Kyriakopoulos, Andreas Miltiadis Seretis, Stefanos Korfias, Stamatios Theocharis, Anna Angelousi

**Affiliations:** 1Department of Neurosurgery, Thriasio General Hospital of Elefsina, 19600 Athens, Greece; 2Department of Pathology, Evaggelismos Hospital, 10676 Athens, Greece; 3Department of Neurology, Medical University of Sofia, 1431 Sofia, Bulgaria; 4Department of Neurosurgery, Evaggelismos Hospital/National and Kapodistrian University of Athens, 10676 Athens, Greece; 51st Department of Pathology, National and Kapodistrian University of Athens, 11527 Athens, Greece; 6Unit of Endocrinology, 1st Department of Internal Medicine, Laiko Hospital, National and Kapodistrian University of Athens, 11527 Athens, Greece

**Keywords:** Cyclin-D1, pituitary, markers, adenoma, Ki-67, prognostic

## Abstract

(1) Background: The aim of the current study is to evaluate the immunohistochemical expression of Ki-67, CD-56, Cyclin-D1 and E-Cadherin in the tissues samples of pituitary adenomas (PAs) and its association with PAs clinical manifestation tumor size, invasiveness and the risk of recurrence. (2) Materials and Methods: Ninety-four patients who underwent endoscope transsphenoidal excision of PAs were included in our study. The immunohistochemical expression of the Cyclin-D1, CD-56, E-Cadherin and Ki-67 markers was analyzed in paraffin-embedded tissue samples. (3) Results: The expression of Cyclin-D1 and Ki-67 index levels was positively correlated with the size (*p* < 0.001, r = 0.56 and *p* < 0.001, r = 0.43, respectively), the recurrence (*p* < 0.001, r = 0.46 and *p* = 0.007 r = 0.3, respectively), the extrasellar extension (*p* < 0.001, r = 0.48 and *p* < 0.001, r = 0.4, respectively) and the cavernous sinus invasion of (*p* < 0.001, r = 0.39 and *p* < 0.001, r = 0.3, respectively). No correlation was found between CD-56 and E-Cadherin expression with the size, the invasiveness and the recurrence of PAs. (4) Conclusion: Cyclin-D1 and Ki-67 are promising immunohistochemical markers in predicting the invasive behavior and recurrence of PAs in contrast to E-Cadherin and CD-56 which did not seem to be associated with PAs behavior post-surgery. However, larger studies are required in order to establish their role in the routine evaluation of PAs.

## 1. Introduction

Pituitary adenomas (PAs) are relatively common endocranial tumors. They represent 10–15% of all intracranial tumors with a prevalence of 0.1% in the overall population [1]. PAs present a wide range of clinical and proliferation behavior. Although PAs usually have benign behavior, significant morbidity can be associated with mass effect and local expansion as well as hormonal deficiency or excess. Non-functional adenomas (NFPAs) represent 14–54% of PAs and have a prevalence of 7–41.3/100.000 population [2]. Functional PAs (FPAs) are classified according to their secretion as prolactin (PRL)—secreting which are the most common representing 45% of all PAs, adrenocorticotropic hormone (ACTH)—secreting (10–15% of all PAs), growth hormone (GH)—secreting (10–15% of all PAs), and more rarely thyroid-stimulating hormone (TSH)—secreting (<1% of all PAs) [3,4]. Follicle—stimulating hormone (FSH) and Luteinizing hormone (LH)—secreting adenomas (gonadotroph adenomas) are usually silent and clinically nonfunctioning and thus difficult to identify them. Approximately 25.2% to 64% of NFPAs are confirmed immunohistochemically as gonadotrophin adenomas [5]. Although in the cases of hormone-secreting pituitary adenomas, hormonal measurements are useful biomarkers for monitoring the disease, in NFPA, there is a need to identify non-invasive biomarkers that are easily accessible by the majority of medical facilities and are cost-effective.

The prediction of the clinical behavior of PAs is complex and challenging. The terminology ‘aggressive’ has been used synonymously with ‘invasive’ when studying PAs. PAs are characterized as aggressive when they present a high risk of recurrence or lack of therapeutic response. Invasive PAs that exhibit high mitotic activity, Ki-67 > 3% or extensive p53 immunoreactivity were classified as ‘atypical adenomas’ by the World Health Organization (WHO) in 2004 [6]. However, in the WHO 2017 classification, the term “atypical adenoma” has been abandoned [7] and aggressive behavior was based on tumor proliferation indexes (mitotic count and Ki-67) and invasion without a specific Ki-67 cut-off value [8,9,10]. The most recent WHO classification in 2022 [11] suggests renaming the anterior lobe tumors—formerly known as PAs—to pituitary neuroendocrine tumors (PitNETs) and classifying them based on cell lineage and cell type [11].

Thus, although several biomarkers such as Ki-67 (cell proliferation marker), pituitary tumor transforming gene (PTTG) (molecular marker for invasiveness), p53 (tumor suppressor protein), fibroblast growth factor receptors (FGFR), and matrix metalloproteinase (MMP), have been investigated over the last years [12,13,14] as potential prognostic parameters, they have not been consistently demonstrated [15].

The aim of the current study is to evaluate the immunohistochemical expression of Ki-67; NCAM (neural cell adhesion molecule) also called CD-56, Cyclin-D1 and E-Cadherin in PAs and to study the possible association of the expression of these markers with the clinical manifestation of PAs, their tumor size and invasiveness as well as with the risk of recurrence.

## 2. Materials and Methods

Our study was approved by the Bioethics Committee of the Medical School of National and Kapodistrian University of Athens (No 142/27.06.2019) and was in accordance with the Helsinki Declaration. Informed consent was obtained from all individual participants included in the study. Additional informed consent for publication was obtained from all participants of the study.

### 2.1. Patients

This retrospective study consisted of 98 patients who underwent endoscopic transsphenoidal excision (eTSS) of the PAs between January 2014 and October 2019 from the same surgical team in the General Hospital of Elefsina “Thriasio”. Patients whose histological reports showed non-adenomatous lesions such as metastatic neoplasms and Rathke’s cysts were excluded from the study (*n* = 4). Thus, the total number of included patients was 94. All patients underwent neurological, ophthalmological and hormonal evaluation before the surgery and during the follow-up period. Follow-up was scheduled every 3 months during the first postoperative year, every 6 months for the next two years and annually for another two years. The mean follow-up period was 2.15 ± 1.4 years. All the available patient information such as demographic data, clinical symptoms and radiological data were collected from the medical records held in our department archive.

### 2.2. Endocrine and Radiological Screening

All patients underwent hormone evaluation, including TSH, FT4, GH, IGF-1, ACTH, cortisol, FSH, LH, E2 and testosterone measurement pre-surgically, as well as during the scheduled follow-up. All the functional hormonal tests were performed at the same laboratory with the use of liquid chromatography-tandem mass spectrometry (LCMS). Imaging evaluation of all PAs was based on magnetic resonance imaging (MRI) with administration of intravenous contrast and MRIs evaluation was performed by the same radiologist team. PAs were divided into two groups according to their size; microadenomas (<10 mm) and macroadenomas (≥10 mm). The invasiveness of PAs was defined as the following: based on Knosp as grade 3 and 4 and based on Hardy’s scale as grade III and IV [16].

### 2.3. Immunohistochemistry

Ninety -four paraffin-embedded tissue samples were cut into 4-μm thick sections, air dried and then placed in an oven at 60 °C overnight. To remove the paraffin wax, the sections were placed in three containers of xylene for 5 min. Afterwards, the sections were placed in two containers of 100% ethanol for 10 min each and in two containers of 95% ethanol for another 10 min each to achieve dehydration. The sections were brought to a boil in 10 mM sodium citrate buffer (pH = 6.0) for ten minutes and then placed in water to avoid drying. The sections were then transferred in blocking buffer [1% horse serum in phosphate-buffered saline (PBS)] to block non-specific staining between the tissue and the antibodies.

All the tissue samples were immunohistochemically examined for the expression of Cyclin-D1, CD-56, E-Cadherin and Ki-67. To detect Cyclin-D1, the sections were incubated with the Clone SP4, rabbit monoclonal antibody (Spring Bioscience) at 1:40 dilution. Clone 123C3, mouse monoclonal antibody at dilution 1:200 was used to detect CD-56; clone NCH-38, mouse monoclonal antibody (Invitrogen Antibodies) at dilution 1:200 was used to detect E-Cadherin. Finally, clone MIB-1, mouse monoclonal antibody at a 1:100 dilution was used to evaluate the expression of Ki-67. Immunoreactivity of proteins, cell-cycle regulators and proliferation markers was determined by manual counting as a proportion of positive cells from a group of 1000 cells. All the slides were separately read by two pathologists who were blind to the clinical and radiological tumor characteristics.

Positive immunochemistry (IHC) expression was scored as the following: absence = 0, mild/weak = 1, moderate = 2, strong/intense = 3. The percentage of the labeled cells was scaled as 0 for 0–5%, 1 for 6–10%, 2 for 11–50%, 3 for 51–80% and 4 for >80% of cells. The final score was calculated by multiplying the intensity score and the percentage of labeled cells: 0 (negative expression), 1–3 (+, weak expression), 4–6 (++, moderate expression), >6 (+++, strong expression) [16].

### 2.4. Statistical Analysis

Categorical variables are presented as absolute values and percentages and data following normal distribution are expressed as mean values with standard deviation. Continuous variables with a normal distribution were compared by unpaired Student’s *t*-test and nonparametric variables with Mann–Whitney test. Categorical variables were compared with the chi-square test and Fisher’s exact test. Two-sided *p*-values < 0.05 were considered statistically significant. The statistical analysis was performed using SPSS (version 25, IBM Corp.).

## 3. Results

### 3.1. Patient’s and Pituitary Adenoma’s Characteristics

We studied 94 patients (55 women) with a mean age of 46.5 (±13.9) years old who underwent eTSS for the treatment of PAs. The clinical, epidemiological and radiological data are shown in Table 1. Sixty-five patients presented with macroadenomas and *n* = 29 patients with microadenomas (mean size = 2 ± 1.1 cm). The most common symptom at diagnosis was visual deficit (*n* = 51/94, 54.3%) followed by headache (*n* = 33/94, 35.1%).

Fifty-seven patients (*n* = 57/94, 60.6%) presented with NFPA; 28 null cell adenomas and 29 gonadotroph adenomas (clinically NFPA); 1 patient presented with panhypopituitarism (1.75%) and 6 (11%) with partial anterior pituitary deficiency pre-operatively. Thirty-seven patients (*n* = 37/94, 39.4%) presented with FPAs; 24 (*n* = 24/37, 65%) with GH-secreting adenomas; 10 patients presented with ACTH-secreting adenomas (*n* = 10/37, 27%) and 3 (*n* = 3/37, 8%) with prolactinomas. Eight out of 24 patients with GH- secreting adenomas (*n* = 8/24, 33.3%) had been treated with long-acting release octreotide for a mean period of 4.2 ± 1.1 months before operation; 4 out of 10 patients with ACTH-secreting adenomas (*n* = 4/10, 40%) had been treated with metyrapone for 4 ± 1.63 months before the surgery whilst all prolactinomas operated were resistant to cabergoline treatment prior to the surgery. The mean hospital stay was 5.2 ± 1.1 days. Total resection was achieved in 67 patients (71.3%). Tumor relapse was observed in 16 patients (*n* = 16/94, 17%), 5 of them (*n* = 5/16, 31.25%) had FPAs (4 GH-secreting, 1 ACTH-secreting) and 11 NFPAs (*n* = 11/16, 68.7%). Fifteen patients (*n* = 15/94, 16%) underwent second operation due to tumor relapse.

### 3.2. Association of Patient’s Clinical and Epidemiological Characteristics with Tumor Characteristics

Men were statistically significantly older at the time of the operation (52.9 vs. 45.6 years, *p* = 0.007) and presented with larger adenomas than women (2.3 vs. 1.8 cm *p* = 0.03). No significant association was found between gender and age with the tumor recurrence ratio (*p* = 0.8, r = 0.16 and *p* = 0.9, respectively). Age was significantly correlated with the size of the PAs (*p* = 0.008, r = 0.3). Additionally, the size and invasiveness of all PAs were significantly correlated with recurrence (*p* = 0.01, r = 0.27 and *p* = 0.04, r = 0.21, respectively).

Thirty-five patients (*n* = 35/94, 37.2%) presented invasive PAs. The rate of invasive of NFPAs was significantly higher (54.4%) compared with the rate of FPAs (10.8%) (*p* = 0.001).

### 3.3. Immunohistopathological Markers of Pituitary Adenomas

#### 3.3.1. Cyclin-D1

Immunohistochemical analysis showed no cytoplasmic staining of paraffin-embedded tissue samples with Cyclin-D1. Positive nuclear staining for Cyclin–D1 was noticed in 79 PAs (*n* = 79/94, 84%) ranging from 5 to 100% of the cells (Figure 1). No significant difference of Cyclin-D1 immunohistochemical expression was found between females (*n* = 45/55, 81.8%) and males (*n* = 34/39, 87.2%) (Table 2).

NFPAs had significantly higher expression of Cyclin-D1 than FPAs (*n* = 50/57, 87.7% vs. *n* = 29/37, 78.4%) (*p* > 0.05). The expression of Cyclin-D1 was positively correlated with the size (*p* < 0.001, r = 0.56) and the volume (mean volume = 5.6 ± 8.2 cm^3^) of all PAs (*p* < 0.001, r = 0.58). Cyclin-D1 expression was also statistically significantly correlated with PA’s recurrence (*p* < 0.001, r = 0.46). Moreover, there was a significant positive correlation between Cyclin-D1 expression and the extrasellar extension of the PAs [(according to Hardy’s classification), *p* < 0.001, r = 0.48)]. In addition, higher expression of Cyclin-D1 were detected in PAs with cavernous sinus invasion [(according to Knosp’s classification), *p* < 0.001, r = 0.39)] (Figure 2).

Regarding NFPAs and FPAs separately, Cyclin-D1 was statistically significantly correlated with their size (*p* = 0.007, r = 0.4 and *p* = 0.005, r = 0.45, respectively) as well as the rate of recurrence (*p* = 0.01, r = 0.32 and *p* < 0.001, r = 0.58, respectively). There was also a significant positive correlation of Cyclin-D1 expression with NFPAs and FPAs extrasellar extension [according to Hardy’s classification, (*p* = 0.03, r = 0.28 and *p* < 0.02, r = 0.38, respectively)]. Additionally, Cyclin-D1 was strongly expressed (expression > 50%) in 78.6% (*n* = 22/28) of the gonadotroph adenomas and in 65.5% (*n* = 19/29) of the null cell adenomas compared to only 20% of corticotroph (*n* = 2/10) and 20.8% of somatotroph (*n* = 5/24) adenomas.

#### 3.3.2. Ki-67 Index Levels

The Ki-67 index value ranged from 0.05 to 10.5% (Figure 1). The Ki-67 index level < 1% was found in 42 (*n* = 42/94, 44.7%) tissue samples; 27 of them were macroadenomas (*n* = 27/42, 64.3%) vs. 15 microadenomas (*n* = 15/42, 35.7%). Eleven tissue samples (*n* = 11/94, 11.7%) had a Ki-67 index level at 1%; 6 (*n* = 6/11, 54.5%) were macroadenomas and 5 (*n* = 5/11, 45.5%) microadenomas. The Ki-67 index level > 1% was found in 41 tissue samples (*n* = 41/94, 43.6%); 32 were macroadenomas (*n* = 32/41, 78%) and 9 microadenomas (*n* = 9/41, 22%) (Table 2).

Ki-67 index levels were significantly positively correlated with the size (*p* < 0.001, r = 0.69), the cavernous sinus invasion of the PAs (based on Knosp’s classification) (*p* < 0.001, r = 0.37), the extrasellar invasion (based of Hardy’s classification) (*p* = < 0.001, r = 0.4) and with PAs’ recurrence (*p* = 0.007 r = 0.3) (Figure 3). In particular, 35 out of 59 (59.3%) PAs classified as Hardy grade I or II had Ki-67 levels < 1%, 8 (13.6%) had Ki-67 level at 1% and 16 (27.1%) had Ki-67 levels > 1%. Seven (20%) out of 35 PAs classified as Hardy grade III or IV had Ki-67 levels < 1%, 3 (8.6%) had Ki-67 levels at 1% and 25 (71.4%) had Ki-67 index levels < 1%. Thirty-five out of 75 (59.3%) PAs classified as grade 0, 1, 2 at Knosp’s scale had Ki-67 index < 1%, 9 (12%) had Ki-67 index at 1% and 26 (34.7%) had Ki-67 index > 1%. Two out of 19 (10.5%) PAs classified as grade 3, 4 at Knosp’s scale had Ki-67 < 1%, 2 (10.5%) had Ki-67 at 1% and 12 (75%) had Ki-67 > 1%.

In NFPAs, the Ki-67 index was significantly positively correlated with their size (*p* = 0.005, r = 0.36) and their recurrence (*p* = 0.03, r = 0.3), as well as of their extrasellar (*p* = 0.01, r = 0.3) and cavernous sinus invasion (*p* = 0.002, r = 0.4). In FPAs, Ki-67 was statistically correlated with their size (*p* = 0.01, r = 0.4) but not with their recurrence (*p* = 0.09, 0.27). Ki-67 was also significantly correlated with the extrasellar invasion of FPAs (*p* = 0.04, 0.3) but not with cavernous sinus invasion (*p* = 0.2, r = 0.2).

Moreover, a statistically significant positive correlation was found between Ki-67 and Cyclin-D1 immunohistochemical expression (*p* = 0.006, r = 0.28). However, the expression of Ki-67 did not differ significantly between FPAs and NFPAs.

#### 3.3.3. CD-56 Expression

CD-56 immunohistochemical expression was positive in 68 paraffin-embedded tissues samples (*n* = 68/94, 72.3%) [39 females (57.4%) and 29 males (42.6%)] ranging from 5% to 100% of cells (Figure 1) (Table 2). Thirty-six samples (52.9%) showed strong expression (+++), 7 (10.3%) showed moderate (++) and 25 (36.7%) showed weak expression (+). Fifty patients (*n* = 50/68, 73.5%) with positive CD-56 expression presented macroadenomas and 18 (*n* = 18/68, 26.5%) microadenomas. No statistically significant correlation was noticed between the expression of CD-56 and the size of total PAs, nor between NFPAs and FPAs [(*p* = 0.5, r = 0.07) for all PAs, (*p* = 0.9, r = 0.02) for NFPAs and (*p* = 0.2, r = 0.2) for FPAs]. Forty-six tissue samples (*n* = 46/68, 67.6%) were Hardy’s I and II (*n* = 46/94, 67.6%) and 22 were Hardy’s III, IV (*n* = 22/94, 32.4%). No significant correlation was found between CD-56 expression and extrasellar invasion of NFPAs or FPAs. Similarly, no correlation was found between CD-56 expression and cavernous sinus invasion or recurrence of NFPAs or FPAs

#### 3.3.4. E-Cadherin

E-Cadherin immunostaining was found positive in 60 tissue samples (*n* = 60/94, 63.8%, 37 females and 23 males) with expression ranging between 5 to 100% (Figure 1). Twenty-four (40%) showed strong expression (+++), 10 (16.7%) showed moderate (++) and twenty-six (43.3%) showed weak (+) expression. Forty-two patients with positive immunohistochemical E-Cadherin expression (*n* = 42/60, 70%) presented with macroadenomas and n = 18 with microadenomas. No correlation was found between the expression of E-Cadherin and the size of the extrasellar invasion, the cavernous sinus invasion or the recurrence in neither NFPAs nor FPAs (Table 2). Nevertheless, we observed that 70.8% of the somatotroph adenomas (*n* = 17/24) presented strong membranous expression of E-Cadherin (>50%) compared to 40% of corticotroph adenomas (*n* = 4/10), 42.8% of gonadotroph adenomas (*n* = 12/28) and 27.6% of null cell adenomas (*n* = 8/29). Additionally, 34 (56%) of PAs presented a total loss of E-Cadherin; 24 (*n* = 24/34, 70.5%) of them had invasive behavior. Moreover, 56.3% of the recurrent adenomas (*n* = 9/16) had a total loss of E-Cadherin and only 12.5% (*n* = 2/16) showed weak to moderate expression.

Forty tissue samples out of these 60 (66.7%) PAs were classified in Hardy’s scale as non-invasive (I, II) and 20 (33.3%) as invasive (Hardy’s scale III, IV. Moreover, the majority (*n* = 49/60, 81.7%) of tissue samples were classified as Knosp’s scale 0, 1, 2 and 11 as Knosp’s scale 3, 4 (*n* = 11/60, 18.3%) 

## 4. Discussion

The majority of the included population presented NFPAs (60.6% with NFPAs vs. 39.4% with FPAs). The size of PAs was associated positively with patients’ age whereas their recurrence with their size and invasiveness. NFPAs had a significantly higher rate of invasion compared with FPAs however no difference was found in the rate of relapse between NFPA and FPAs. The immunohistochemical analysis seems to contribute to the prediction of PAs behavior since Cyclin-D1 and Ki-67% were both significantly positively correlated with PAs recurrence and invasion although Ki-67% was a statistically significant biomarker only for NFPAs relapse. CD-56 and E-Cadherin immunohistochemical expression were not significantly correlated with PAs relapse or invasion.

Cyclin-D1 is an important cell cycle regulator and plays an important role as an oncoprotein in tumor proliferation. High levels of Cyclin-D1 are required to sustain tumor growth [17,18]. In our study, Cyclin-D1 was positively correlated with size, recurrence and with the invasiveness (according to Hardy’s and Knosp’s scale) of PAs. Similarly to our results, a previous [19] retrospective study including 74 PA samples reported a positive correlation of Cyclin-D1 with the size, the suprasellar and cavernous sinus extension [18]. Another larger retrospective study, including 297 patients reported a positive correlation between Cyclin-D1 and the recurrence of PAs [20]. Interestingly, in our study, Cyclin-D1 was particularly strongly expressed in most of the gonadotroph adenomas, an observation shared also with the study of Hewedi et al., which had also reported higher expression of Cyclin-D1 in gonathotroph and null cell adenomas [21].

Ki-67 is a nuclear antigen recognized by the monoclonal antibody MIB-1 associated with cellular proliferation [22]. In our study, Ki-67 index levels were positively correlated with the size, extrasellar extension, cavernous sinus invasion and PAs recurrence. A previous retrospective analysis of 55 patients with PAs, reported a positive correlation between Ki-67 and the size of the PAs [23]. A retrospective study by Glebauciene et al. reported a positive significant correlation between Ki-67 expression and the PA invasion but not with Hardy’s scale or Knosp’s scale [24]. This could be due to the small sample of patients since larger studies [25,26] have shown a positive significant correlation between Ki-67 index levels with the invasiveness of PAs (following the Knosp scale) [25] and their recurrence [26]. Finally, the Ki-67 index was also positively correlated with Cyclin-D1 indicating a constant number of cells that have entered the cell cycle continue to proliferate. These results are in agreement with several studies reporting a correlation between cyclins expression, cell proliferation and tumor progression [27,28]. Hewedi et al. studied 199 PAs and also found a positive correlation between these two markers [21].

CD-56 is a homophilic binding glycoprotein which expressed on the surface of L cells and muscle fibers and plays an important role in the proliferation and differentiation of cells. As a biomarker, CD-56 can be also expressed in normal neuroendocrine cells and neuroendocrine neoplasms, and thus, it is considered a potential neuroendocrine marker whereas the loss of its expression has been associated with increased metastatic risk, progression of malignant neoplasms such as myeloma, myeloid leukemia, pheochromocytoma, cholangiocarcinoma and paraganglioma [29,30]. Regarding adenohypophyseal cells, considered neuroendocrine cells expressing neuroendocrine proteins such as synaptophysin, chromogranin A and CD-56 [31], the existing data from the literature although limited have demonstrated that CD-56 expression does not differ between normal pituitary gland cells and PA cells [32,33]. Furthermore, CD-56 does not seem to be related to either PAs proliferation or its invasiveness [32,34,35]. Indeed, in our study CD-56 expression was not significantly correlated with any imaging characteristic of PA (size, extrasellar or cavernous sinus invasion) nor with recurrence.

E-Cadherin (Cadherin of epithelial origin) is a cell adhesion protein encoded by the gene *Cadherin-1* (*CDH1*). The loss of its expression on the cell surface in the immunochemical analysis is associated with invasiveness, metastasis and bad prognosis in several malignancies such as breast and ovarian cancer [36]. Similarly, adequate cell-to-cell adhesion is crucial for the epithelial phenotype of pituitary cells. However, there are some controversies regarding E-Cadherin’s role in PA’s growth and invasiveness [37,38]. Some authors have reported that loss of E-Cadherin was associated with the invasiveness and dedifferentiated phenotype of GH-secreting adenomas, also presenting a negative correlation with tumor size and positive correlation with response to somatostatin analogs [37] whereas others have failed to show any correlation between E-Cadherin and tumor size or invasiveness [29]. In our study, we observed that 70.8% of the GH-secreting adenomas (*n* = 17/24) presented strong membranous expression of E-Cadherin (>50%). Additionally, 68.75% of the relapsed PAs showed weak to moderate or total loss of E-Cadherin expression. However, we found no statistically significant correlation between E-Cadherin and size, extrasellar invasion, cavernous sinus invasion and recurrence similarly to previously published data [33,39,40,41]. In one study, [17] including 91 cases, E-Cadherin’s protein expression was positively correlated with tumor invasiveness in FPAs [39]. In another study, including 83 GH-secreting adenomas, E-Cadherin presented strong membranous expression (>50%) in 80% of GH-secreting adenomas and a negative correlation with tumor size whereas, in another study including 83 adenomas [42], low or absence of E-Cadherin expression was correlated with tumor invasiveness.

In our study, we observed that E-Cadherin presented strong membranous expression in the majority of somatotroph adenomas (>70%) compared to only 27.6% of null cell adenomas whereas Cyclin-D1 was strongly expressed (expression > 50%) in 78.6 % of the gonadotroph adenomas and in 65.5% of the null cell adenomas compared to only 20% of corticotroph and somatotroph adenomas. Although, E-Cadherin expression has been already studied in patients with acromegaly and found that low expression has been associated with the worst response to somatostatin analogs [36], data on the possible association of PAs secretory profile with the expression of specific biomarkers such as Cyclin-D1 and E-Cadherin are rare and unclear [21,43,44,45].

The main limitation of our study is the relatively small number of patients and of the paraffin-embedded blocks available for immunohistochemistry, as well as the short follow-up, especially regarding the evaluation of PA's recurrence. However, our results are in agreement with the majority of the recently published data. Moreover, we have included clinical parameters of the studied population as well as radiological characteristics of the PAs, which we tried to connect and explain based on the (immune) histopathological findings when possible.

## 5. Conclusions

Cyclin -D1 and Ki-67 are promising immunohistochemical markers in predicting the invasive behavior and recurrence of PAs whereas E-Cadherin and CD-56 did not seem to be associated with PA behavior post-surgery although more than 60% of the relapsed PAs presented low or null expression of E-Cadherin. The size and invasion of the tumor based on MRI and Hardy’s and Knosp’s criteria were both significantly correlated with recurrence. In addition, Cyclin-D1 and E-Cadherin expression seem to differ based on the neurosecretory profile of PA cells. Thus, combing immunochemistry with imaging characteristics could help in predicting PA behavior. Additional studies with larger samples are required to identify new predictive factors for PAs.

## Figures and Tables

**Figure 1 diagnostics-12-02413-f001:**
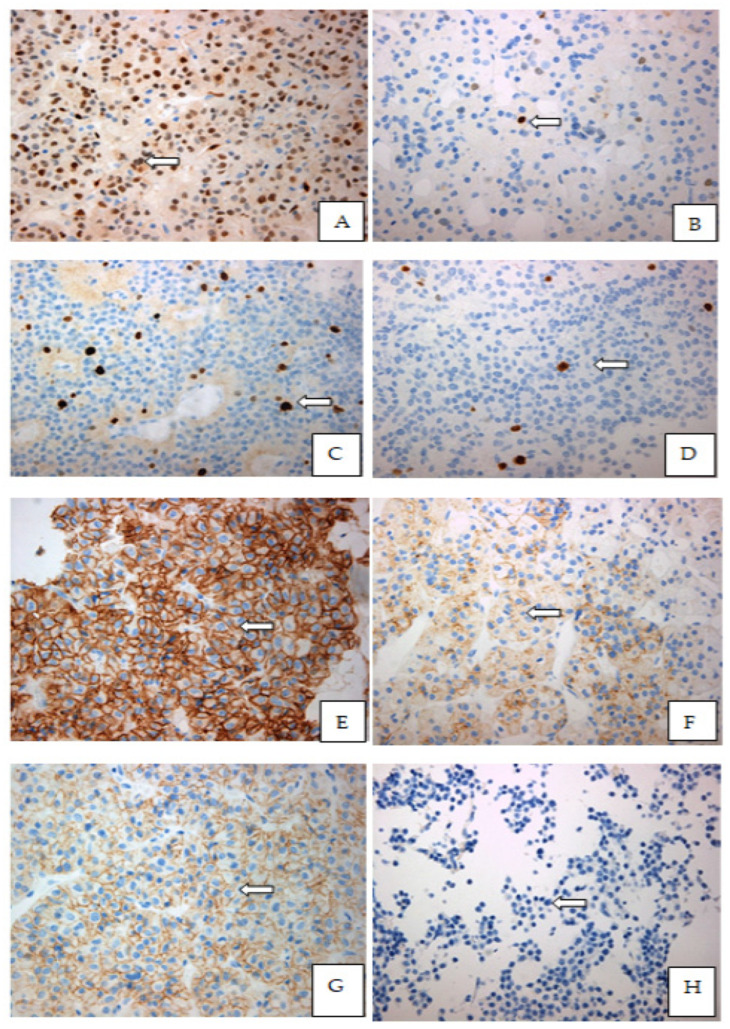
Immunohistochemical expression patterns of the Cyclin D1, E-Cadherin, CD56, Ki-67% in paraffin-embedded tissues of pituitary adenomas: (**A**) Strong nuclear expression of cyclin D1 in the majority of the neoplastic cells (9550.13 × 400), of a gonadotroph adenoma GH-secreting pituitary adenomas; (**B**) Moderate nuclear expression of Cyclin-D1 in few dispersed pituitary cells from a GH-secreting pituitary adenoma (9359.13 × 400); (**C**) Moderate proliferative index Ki-67 in hotspot, in a non-functioning pituitary macroadenoma; (**D**) Low proliferative index Ki-67% in hotspot in a somatotroph adenoma; (**E**) Strong and complete membranous expression of CD56 (9550.13 × 400) in a non-functioning pituitary adenoma; (**F**) Mild to moderate incomplete membranous expression of CD56 (7115.15 × 400) in a ACTH-secreting pituitary adenoma; (**G**) Strong and complete membranous expression of E-Cadherin (9550.13 × 400) in the majority of the cells in a non-functional pituitary adenoma; (**H**) Complete loss of membranous expression of E-Cadherin (7163.13 × 400) in a GH-secreting pituitary adenoma.

**Figure 2 diagnostics-12-02413-f002:**
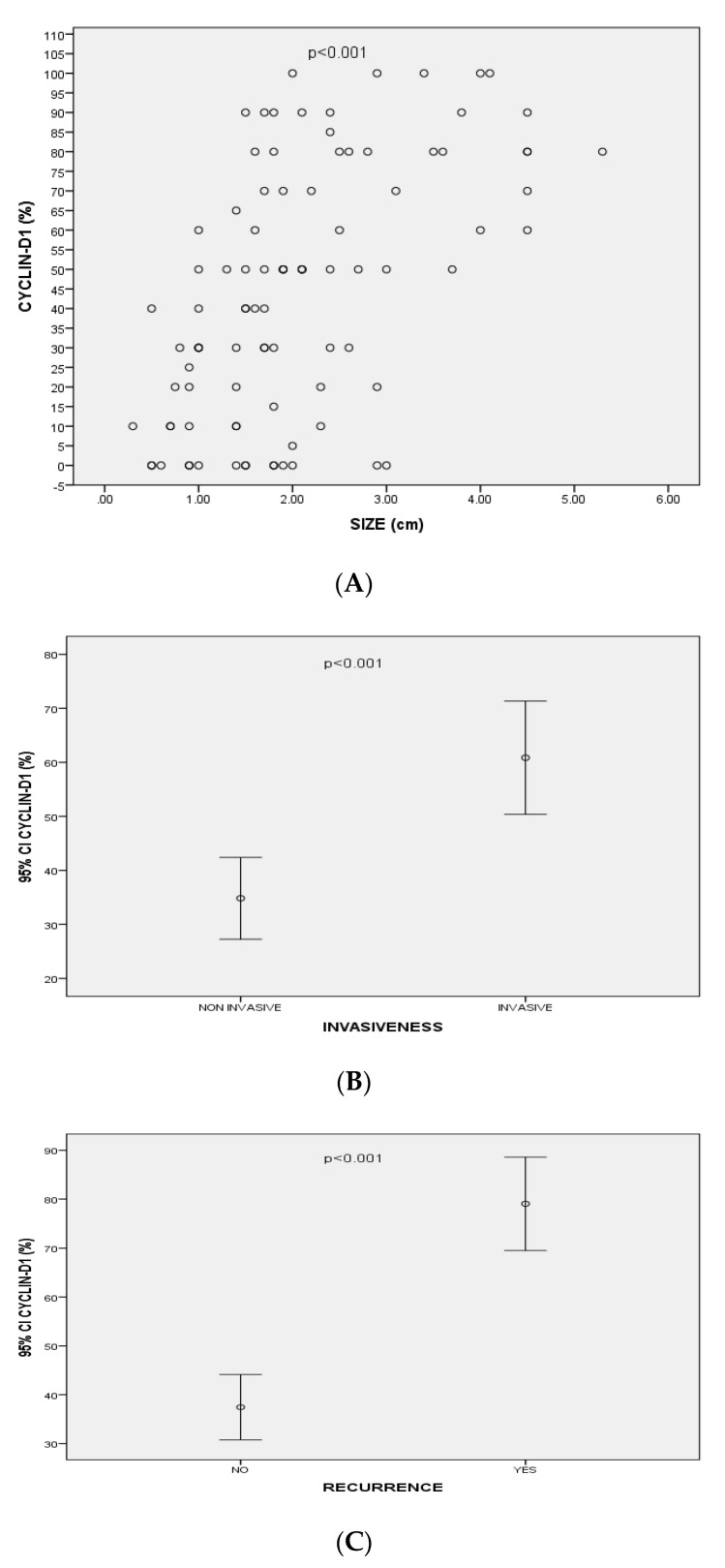
(**A**) Correlation between Cyclin-D1 expression and size of pituitary adenomas (*p* < 0.001, r = 0.56); (**B**) Mean values of Cyclin-D1 expression (95% CI) associated with invasiveness of pituitary adenomas; (**C**) Mean values of Cyclin-D1 expression (95% CI) associated with recurrence of pituitary adenomas.

**Figure 3 diagnostics-12-02413-f003:**
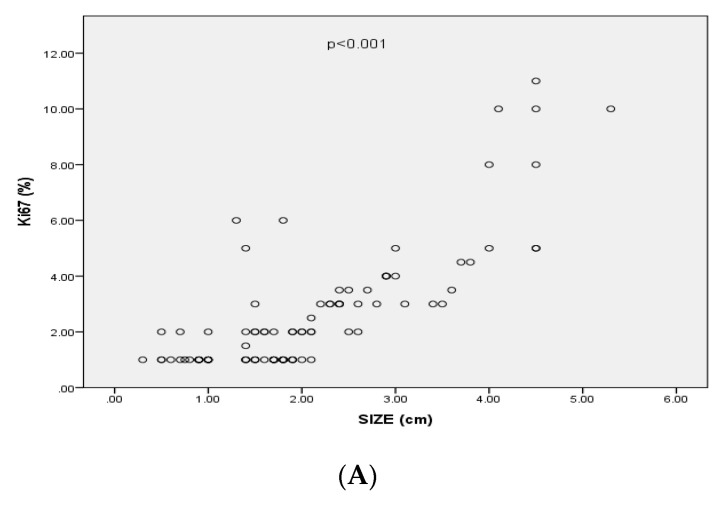
(**A**) Correlation between Ki-67 expression and size of pituitary adenomas (*p* < 0.001, r= 0.69); (**B**) Mean values of Ki-67 expression (95%CI) associated with invasiveness of pituitary adenomas; (**C**) Mean values of Ki-67 expression (95%CI) associated with recurrence of pituitary adenomas.

**Table 1 diagnostics-12-02413-t001:** Baseline characteristics of the included patients.

Variables	Frequency or Mean ± SD
Age (yrs) Diagnose age	48.6 ± 13.5 46.5 ± 13.9
Gender	
Male Female	39/94 (41.5%) 55/94 (58.5%)
Tumor Diameter (cm)	2.1 ± 1.1
Tumor Size	
Macroadenoma (≥1 cm) Microadenoma (<1 cm)	65/94 (69.1%) 29/94 (30.9%)
Adenoma subtype	
GH ACTH PRL Non-secreting (non-functioning)	24/94 (25.5%) 10/94 (10.6%) 3/94 (3.2%) 57/94 (60.6%)
Symptoms	
Apoplexy Visual Deficit Headache	6/94 (6.4%) 51/94 (54.3%) 33/94 (35.1%)
Invasiveness	
Non-invasive Invasive	59/94 (62.8%) 35/94 (37.2%)
Resection	
Total Partial	67/94 (71.3%) 27/94 (28.7%)
Resection rate	
66–80% 81–95% >96%	7/94 (7.4%) 21/94 (22.3%) 66/94 (70.2%)
Imaging Recurrence	16/94 (17%)
Functional Recurrence	5/37 (13.5%)
Re-operation	15/94 (16%)

Abbreviations: GH: growth hormone; ACTH: adrenocorticotropic hormone; PRL: prolactin.

**Table 2 diagnostics-12-02413-t002:** Expression of Cyclin-D1, Ki-67, CD-56, E-Cadherin (NFPAs and FNPAs).

	Cyclin-D1			Ki-67					CD-56			E-Cadherin		
				<1%	1%	>1%								
	N (%)	Statistical Significance (*p*)	Correlation (r)	N (%)			Statistical Significance (*p*)	Correlation (r)	N (%)	Statistical Significance (*p*)	Correlation (r)	N (%)	Statistical Significance (*p*)	Correlation (r)
Size (macro vs. microadenoma)	56 (70.9%) vs. 23 (29.1%)	<0.001	0.56	27 (41.5%) vs. 15 (51.7%)	6 (9.2%) vs. 5 (17.2%)	32 (49.2%) vs. 9 (31%)	<0.001	0.69	50 (73.5%) vs. 18 (26.5%)	0.5	0.07	42 (70%) vs. 18 (30%)	0.4	−0.1
Extrasellar Invasion (Hardy I, II vs. III, IV)	46 (58.2%) vs. 33 (41.8%)	<0.001	0.48	35 (59.3%) vs. 7 (20%)	8(13.6%) vs. 3(8.6%)	16 (27.1%) vs. 25 (71.4%)	<0.001	0.4	46 (67.6%) vs. 22 (32.4%)	0.26	−0.1	40 (66.7%) vs. 20 (33.3%)	0.9	−0.002
Cavernous Sinus Invasion (Knosp 0, 1, 2 vs. 3, 4)	61 (77.2%) vs. 18 (22.8%)	<0.001	0.39	40(53.3%) vs. 2(10.5%)	9(12%) vs. 2(10.5%)	26(34.7%) vs. 15(78.9%)	<0.001	0.37	54(79.4%) vs. 14(20.6%)	0.7	0.03	49(81.7%) vs. 11(18.3%)	0.7	−0.03
Recurrence	16/16 (100%)	<0.001	0.46	3 (18.8%)	1 (6.2%)	12 (75%)	0.007	0.3	15/16 (93.8%)	0.06	0.2	12/16 (75%)	0.5	−0.05

## Data Availability

The data that support the findings of this study are available from the corresponding author [P.S.].

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
