# Peer review of "Prognostic Factors for Invasiveness and Recurrence of Pituitary Adenomas: A Series of 94 Patients"

_diagnostics, 2022, doi:10.3390/diagnostics12102413_

Round 1

Reviewer 1 Report

In this manuscript, the authors have evaluated the association between the expression of Ki-67, CD-56(NCAM), Cyclin-D1 and E-Cadherin in PAs with clinical parameters. They found that Ki-67 and Cyclin-D1 can be used to predict invasive behavior and recurrence of PAs. Though, the research design and methodology are appropriate, the authors should address the following point to improve the manuscript.

1. The novelty of the study should be clearly emphasized as the prognostic value of Cyclin-D1 and Ki-67 was demonstrated in a considerable number of publications.

2. What is the rationale to select E-cadherin and CD56?

3. The gene level of the four markers should be investigated.

4. The representative immunohistochemical images of all four markers are needed.

Author Response

REVIEWER 1

In this manuscript, the authors have evaluated the association between the expression of Ki-67, CD-56(NCAM), Cyclin-D1 and E-Cadherin in PAs with clinical parameters. They found that Ki-67 and Cyclin-D1 can be used to predict invasive behavior and recurrence of PAs. Though, the research design and methodology are appropriate, the authors should address the following point to improve the manuscript.

1.The novelty of the study should be clearly emphasized as the prognostic value of Cyclin-D1 and Ki-67 was demonstrated in a considerable number of publications.

Answer: The Reviewer raises the question of novelty. Indeed there are numerous studies in the literature, showing the association of Cyclin-D1 [18-21] and Ki67 [22-26] with size, invasiveness and recurrence. However this is not the case for E-Cadherin and CD-56 since current data in the literature are more controversial. We note the following observations which to our knowledge haven’t been widely investigated and which they could motivate further analysis. The following data have been now incorporated more clearly in the discussion part (p.11, lines 329-337): «In our study we observed that E-Cadherin presented strong membranous ex-pression in the majority of somatotroph adenomas (>70%) compared to only 27.6% of of null cell adenomas whereas Cyclin-D1 was strongly expressed (expression>50%) in 78.6 % of the gonadotroph adenomas and in 65.5% of the null cell adenomas compared to only 20% of corticotroph and somatotroph adenomas. Although, E-Cadherin expression has been already studied  in patients with acromegaly and found that low expression has been associated with worst response to somatostatin analogues [36], data on the possible association of PAs secretory profile with the expression of specific biomarkers such as Cyclin-D1 and E-Cadherin are rare and unclear [21,43, 44, 45]».

  1. What is the rationale to select E-cadherin and CD56?

Answer: Our purpose was to choose immunohistopathological markers of routine already used in other tumors and to study their expression and their predictive role regarding the clinical behavior of PA. E-cadherin was one of them and its choice was further supported by the fact that there were several controversies regarding its role in PA’s growth and invasiveness in the existing literature [36, 37]. We have now added the following data (p.11, lines 311-321): “Similarly, adequate cell-to-cell adhesion is crucial for the epithelial phenotype of pituitary cells. However, there are some controversies regarding E-Cadherin’s role in PA’s growth and invasiveness [37, 38]. Some authors have reported that loss of E-Cadherin was associated with the invasiveness and dedifferentiated phenotype of GH-secreting adenomas, presenting also negative correlation with tumor size and positive correlation with response to somatostatin analogues [37] whereas others have failed to show any correlation between E-Cadherin and tumor size or invasiveness [29]. In our study, we observed that 70.8% of the GH-secreting adenomas (n=17/24) presented strong membranous expression of E-Cadherin (>50%). Additionally, 68.75% of the relapsed PAs showed weak to moderate or total loss of E-Cadherin expression”.

Concerning CD-56, we have now added the following data (p.10, lines 296-305): “As biomarker, CD-56 can be also expressed in normal neuroendocrine cells and neuroendocrine neoplasms and thus it is considered as a potential neuroendocrine marker whereas the loss of its expression it has been associated with increase metastatic risk, progression of malignant neoplasms such as myeloma, myeloid leukemia, pheochro-mocytoma, cholangiocarcinoma and paraganglioma [29, 30]. Regarding adenohypophyseal cells, considered as neuroendocrine cells expressing neuroendocrine proteins such as synaptophysin, chromogranin A and CD-56 [31], the existing data from the literature although limited have demonstrated that CD-56 expression doesn't differ between normal pituitary gland cells and PA cells,  [32, 33]. Furthermore, CD-56 does not seem to be related to neither PAs proliferation nor its invasiveness [32, 34, 35]”. The existing data of CD-56 on PA were limited and thus we decided to include also this marker CD-56 to our study.

  1. The gene level of the four markers should be investigated.

Answer: We thank the Reviewer for this very interesting suggestion. Unfortunately, this is a limitation to this project. Genes’ analyses in mRNA level would be difficult since most of these paraffin embedded tissues were relatively old with insufficient material in most cases for further experiments and thus DNA or RNA extraction could be technically difficult requiring furthermore extra time and cost.

  1. The representative immunohistochemical images of all four markers are needed.

Answer: We thank the Reviewer for notifying us for this omission. Indeed, we had included as supplementary file a figure with the immunohistochemical images of all four markers. We have moved this figure now and incorporate it in the main as Fig 1 (p. 13).

Reviewer 2 Report

This manuscript that submitted by Petros et al. aiming to evaluated if the biomarkers of Ki-67, CD56(NCAM), Cyclin-D1 and E-Cadherin could serve as Prognostic factors in pituitary adenomas (PA). The authors provided the evidences that Ki-67 and Cyclin D1 had significant correlation with tumor size, but CD56 and E-cadherin didn't have any significant correlation in PA. There are some suggestions for the authors

Major concerns

1. Ki-67 and Cyclin D1 are well-documented correlation with tumor size, but using these two biomarkers as prediction factors in PA lacks the novelty in present study. It seems moving well-documented biomarkers from other types of tumor on PA. This part of results are most likely expected.

2. Comparing with well-documented prognostic factors, what is the rational reasons the authors deciding to evaluate the correlation of CD56 and E-cadherin in PA. In my opinion, the E-cadherin plays ambiguous role during tumor progression, the high expression level may associated with tumor growth, on the other hand, the low expression level may associated with metastasis. The CD56 plays an important role in physiological function of neuron system, the authors should provide the relevant refs. on how to make interpretions of CD56 level in PA. This part results lack convincing presentation.

3. Do authors have other candidates for evaluating suitable prognostic factors on PA?

Author Response

REVIEWER 2

This manuscript that submitted by Petros et al. aiming to evaluated if the biomarkers of Ki-67, CD56 (NCAM), Cyclin-D1 and E-Cadherin could serve as Prognostic factors in pituitary adenomas (PA). The authors provided the evidences that Ki-67 and Cyclin D1 had significant correlation with tumor size, but CD56 and E-cadherin did not have any significant correlation in PA. There are some suggestions for the authors

Major concerns

  1. Ki-67 and Cyclin D1 are well-documented correlation with tumor size, but using these two biomarkers as prediction factors in PA lacks the novelty in present study. It seems moving well-documented biomarkers from other types of tumor on PA. This part of results is most likely expected.

Answer: We thank the Reviewer for raising the issue of novelty. Indeed there are numerous studies showing that Cyclin-D1 [18-21] and Ki67 [22-26] were associated with size, invasiveness and recurrence. However, this is not the case for E-Cadherin and CD56 since data are more controversial regarding these markers. We note the following observations which to our knowledge haven’t been widely investigated and which they could motivate further analysis. The following data have been now incorporated more clearly in the discussion part (p.11, lines 329-337): “In our study we observed that E-Cadherin presented strong membranous expression in the majority of somatotroph adenomas (>70%) compared to only 27.6% of of null cell adenomas whereas Cyclin-D1 was strongly expressed (expression>50%) in 78.6 % of the gonadotroph adenomas and in 65.5% of the null cell adenomas compared to only 20% of corticotroph and somatotroph adenomas. Although, E-Cadherin expression has been already studied  in patients with acromegaly and found that low expression has been associated with worst response to somatostatin analogues [36], data on the possible association of PAs secretory profile with the expression of specific biomarkers such as Cyclin-D1 and E-Cadherin are rare and unclear [21,43, 44, 45].”

  1. Comparing with well-documented prognostic factors, what are the rational reasons the authors deciding to evaluate the correlation of CD5-6 and E-Cadherin in PA? In my opinion, the E-cadherin plays ambiguous role during tumor progression, the high expression level may associated with tumor growth, on the other hand, the low expression level may associated with metastasis. The CD-56 plays an important role in physiological function of neuron system, the authors should provide the relevant refs. on how to make interpretations of CD-56 level in PA. This part results lack convincing presentation.

Answer: We totally agree with the Reviewer’s statement regarding CD-56 and E-Cadherin in PA. Actually, our purpose was to choose immunohistopathological markers of routine already used in other tumors and to study their expression and their predictive role regarding the behavior of PA. E-cadherin was one of them and its choice was further supported by the fact that there were several controversies regarding its role in PA’s growth and invasiveness in the existing literature [36, 37]. We have now added the following data (p.11, lines 311-321): “Similarly, adequate cell-to-cell adhesion is crucial for the epithelial phenotype of pituitary cells. However, there are some controversies regarding E-Cadherin’s role in PA’s growth and invasiveness [37, 38]. Some authors have reported that loss of E-Cadherin was associated with the invasiveness and dedifferentiated phenotype of GH-secreting adenomas, presenting also negative correlation with tumor size and positive correlation with response to somatostatin analogues [37] whereas others have failed to show any correlation between E-Cadherin and tumor size or invasiveness [29]. In our study, we observed that 70.8% of the GH-secreting adenomas (n=17/24) presented strong membranous expression of E-Cadherin (>50%). Additionally, 68.75% of the relapsed PAs showed weak to moderate or total loss of E-Cadherin expression”.

Concerning CD-56, we have now added the following data (p.10, lines 296-305): “As biomarker, CD-56 can be also expressed in normal neuroendocrine cells and neuroendocrine neoplasms and thus it is considered as a potential neuroendocrine marker whereas the loss of its expression it has been associated with increase metastatic risk, progression of malignant neoplasms such as myeloma, myeloid leukemia, pheochromocytoma, cholangiocarcinoma and paraganglioma [29, 30]. Regarding adenohypophyseal cells, considered as neuroendocrine cells expressing neuroendocrine proteins such as synaptophysin, chromogranin A and CD-56 [31], the existing data from the literature although limited have demonstrated that CD-56 expression doesn't differ between normal pituitary gland cells and PA cells,  [32, 33]. Furthermore, CD-56 does not seem to be related to neither PAs proliferation nor its invasiveness [32, 34, 35]”. The existing data of CD-56 on PA were limited and thus we decided to include also this marker CD-56 to our study.

  1. Do authors have other candidates for evaluating suitable prognostic factors on PA?

Answer: We thank the Reviewer for this suggestion. Indeed, it would be useful to evaluate more prognostic markers. Unfortunately, insufficient tissue specimen and limited budget did not allow further analyses. Our rational was to analyze markers, which could be easily used as a “routine” by the majority of pathological  laboratories.  

Reviewer 3 Report

The authors presented the prognostic factors for invasiveness and recurrence of pituitary 2 adenomas in a series of 94 patients. The paper mainly used endocrine and radiological screening and immunohistochemistry methods to achieve the results. Although a table illustrating the results, the paper lacks using graphs to show the significant value of the results. Also when referring to immunohistochemistry, there should be images included in the results to support the claims.    

Author Response

REVIEWER 3

The authors presented the prognostic factors for invasiveness and recurrence of pituitary adenomas in a series of 94 patients. The paper mainly used endocrine and radiological screening and immunohistochemistry methods to achieve the results. Although a table illustrating the results, the paper lacks using graphs to show the significant value of the results. Also when referring to immunohistochemistry, there should be images included in the results to support the claims.   

Answer: We thank the Reviewer for this suggestion, which could ameliorate the quality of our paper. We have now added Graphs (Fig.2, 3) (p.14-15) showing correlation of Cyclin-D1 and Ki-67 with size, invasiveness and recurrence. Indeed, we had included as supplementary file a figure with the immunohistochemical images of all four markers. We have now moved this figure and incorporate it in the main as Fig 1 (p.13)

Reviewer 4 Report

The authors studied PAs retrospectively. They tried to find the relationship between immunohistochemical study and tumor behaviors. I have some concern as follows.

Line 15: “CD-56(NCAM)”. Don’t write NCAM in abstract. CD-56 is enough.

Line 21: “Cyclin-D1and”. There must be a space before “and”.

Line 26: “Cyclin -D1”. The hyphen is at wrong side like line 321. The true writing is “Cyclin-D1”

Line 27: “E-cadherin”. Previously it is written like “E-Cadherin”. Please convert “E-cadherin” to “E-Cadherin” throughout the paper.

Line 68: “CD-56(NCAM)”. Yes, CD-56 is neural cell adhesion molecule. But, only abbreviation NCAM is not enough. It is better to give full phrase and later use abbreviation. Moreover there must be space after CD-56, before paranthesis.

Line 81: What are the pathologies of non adenomatous lesions, please declare.

Line 91-93: The authors rewrite the full phrase of the hormones. In fact they give them in introduction. This is not necessary. Please revise this.

Line 117: “mouse monoclonal antibody (DAKO)”. This is repeat. Is it necessary to repeat DAKO?

Line 179: ”cyclin D1” in little letters. Previously or later it is written as “Cyclin-D1”. Please write Cyclin-D1 for every usage of the marker, such as lines 190, 262.

Line 255: “E-CADHERIN”. In top description it is better to write “E-Cadherin”.

Line 273: “of PAs (p=0.029)”. This is discussion part and it is not necessary to write p values. This can make the readers tired. Please delete all p values thgroutht the dicussion part, if not only too much necessaty (lines 279,281,283,284,291).

Line 288: “proliferate These”. The dot has been forgotten after “proliferate”.

Line 299: “Cadherin-1(CDH1)”. There must be space after Cadherin-1 “Cadherin-1 (CDH1”.

Line 304: “Lipe et al.[35]”. There must be space after al. like that “Lipe et al. [35]”. And the authors should check all grammer problems like that all over the paper. Moreover There are no paranthesis closing sign and dot at the end of the sentences. The true format is “Lipe et al. [35]).”.

Author Response

-REVIEWER 4  

The authors studied PAs retrospectively. They tried to find the relationship between immunohistochemical study and tumor behaviors. I have some concern as follows.

-Line 15: “CD-56(NCAM)”. Don’t write NCAM in abstract. CD-56 is enough.

Answer: We thank the Reviewer for this correction. We have now corrected it.

-Line 21: “Cyclin-D1and”. There must be a space before “and”.

Answer: We thank the Reviewer for this correction. We have now corrected it.

-Line 26: “Cyclin -D1”. The hyphen is at wrong side like line 321. The true writing is “Cyclin-D1”

Answer: We thank the Reviewer for this correction. We have now corrected it.

-Line 27: “E-cadherin”. Previously it is written like “E-Cadherin”. Please convert “E-cadherin” to “E-Cadherin” throughout the paper.

Answer: We thank the Reviewer for this correction. We have now corrected it.

-Line 68: “CD-56(NCAM)”. Yes, CD-56 is neural cell adhesion molecule. But, only abbreviation NCAM is not enough. It is better to give full phrase and later use abbreviation. Moreover there must be space after CD-56, before paranthesis.

Answer: We thank the Reviewer for this correction. We have now corrected it both comments.

-Line 81: What are the pathologies of non adenomatous lesions, please declare.

Answer: We have now added some pathologies of non adenomatous lesions (metastatic neoplasms and Rathke’s cysts) (p. 2, lines 82-83).

-Line 91-93: The authors rewrite the full phrase of the hormones. In fact, they give them in introduction. This is not necessary. Please revise this.

Answer: We thank the Reviewer for this correction. We have now revised it (p.3, lines 91-92).

-Line 117: “mouse monoclonal antibody (DAKO)”. This is repeat. Is it necessary to repeat DAKO? Answer: We thank the Reviewer for this correction. We have now corrected the repeat.

-Line 179: ”Cyclin D1” in little letters. Previously or later it is written as “Cyclin-D1”. Please write Cyclin-D1 for every usage of the marker, such as lines 190, 262.

Answer: We thank the Reviewer for this correction. We have now corrected it throughout the manuscript.

-Line 255: “E-CADHERIN”. In top description it is better to write “E-Cadherin”.

Answer: We thank the Reviewer for this correction. We have now corrected it.

-Line 273: “of PAs (p=0.029)”. This is discussion part and it is not necessary to write p values. This can make the readers tired. Please delete all p values through the discussion part, if not only too much necessary (lines 279,281,283,284,291).

Answer: We thank the Reviewer for this correction. We have now deleted the p values from the discussion part.

-Line 288: “proliferate These”. The dot has been forgotten after “proliferate”.

Answer: We thank the Reviewer for this correction. We have now corrected it.

-Line 299: “Cadherin-1(CDH1)”. There must be space after Cadherin-1 “Cadherin-1 (CDH1)”.

Answer: We thank the Reviewer for this correction. We have now corrected it.

-Line 304: “Lipe et al.[35]”. There must be space after al. like that “Lipe et al. [35]”. And the authors should check all grammer problems like that all over the paper. Moreover there are no paranthesis closing sign and dot at the end of the sentences. The true format is “Lipe et al. [35]).”.

Answer: We thank the Reviewer for this correction. We have now corrected it.

Round 2

Reviewer 1 Report

The authors have addressed all the comments and the manuscript can be accepted for publication.

Author Response

The authors have addressed all the comments and the manuscript can be accepted for publication.

Answer: We thank the Reviewer for his suggestions which helped us to ameliorate our manuscript . We are glad that we answered to all his comments .

Reviewer 2 Report

The authors responded all questions and make this manuscript becoming better. I approve this manuscript publishing.

Author Response

The authors responded all questions and make this manuscript becoming better. I approve this manuscript publishing.

Answer:We thank the Reviewer for his suggestions which helped us to ameliorate our manuscript . We are glad that we answered to all his comments.

Reviewer 3 Report

The authors addressed the reviewers comments. However, to establish Immunohistochemistry as a prognostic assay, the results should be also supported by other assays including western blot analysis and imaging techniques. I would suggest this paper needs more confirmatory experiments.

Author Response

The authors addressed the reviewer’s comments. However, to establish Immunohistochemistry as a prognostic assay, the results should be also supported by other assays including western blot analysis and imaging techniques. I would suggest this paper needs more confirmatory experiments.

Answer: We thank the Reviewer for this very interesting suggestion.  Indeed, a western blot analysis could support further our immunohistochemical results. However, technically an additional experimental analysis including Western blot or any other molecular analysis would be difficult because most of our paraffin embedded tissues were relatively old (>5years) with insufficient material in most cases for further experiments. Thus, we have decided similarly to previous studies (21, 23, 33, 34) to proceed with immunohistochemical analysis solely of our available tissue samples.

Reviewer 4 Report

The authors made all corrections that I have mentioned.  Only p values are still there at the lines 298 and 299. It is better to delete them.

Author Response

The authors made all corrections that I have mentioned.  Only p values are still there at the lines 298 and 299. It is better to delete them.

Answer: We thank the reviewer for his comment. We have now deleted the p values.

Round 3

Reviewer 3 Report

Since the additional experiments are not possible on the used samples, the current format could be acceptable.